**Data availability statement:** Data Availability and access The datasets analyzed during the current study are available from the following public domain resources:

# UFM: Unified feature matching pre-training with multi-modal image assistants

**Yide Di**[1,3], **Yun Liao**[2,3]*, **Hao Zhou**[3], **Kaijun Zhu**[3], **Qing Duan**[2], **Junhui Liu**[2], **Mingyu Lu**[1]*

**1** Information Science and Technology College, Dalian Maritime University, Dalian, China, **2** National Pilot School of Software, Yunnan University, Kunming, Yunnan, China, **3** Yunnan Lanyi Network Technology Co., Kunming, China

* LiaoYun@ynu.edu.cn (YL); dlmuitrec@163.com (ML)

## Abstract

Image feature matching, a foundational task in computer vision, remains challenging for multimodal image applications, often necessitating intricate training on specific datasets. In this paper, we introduce a Unified Feature Matching pre-trained model (UFM) designed to address feature matching challenges across a wide spectrum of modal images. We present Multimodal Image Assistant (MIA) transformers, finely tunable structures adept at handling diverse feature matching problems. UFM exhibits versatility in addressing both feature matching tasks within the same modal and those across different modals. Additionally, we propose a data augmentation algorithm and a staged pre-training strategy to effectively tackle challenges arising from sparse data in specific modals and imbalanced modal datasets. Experimental results demonstrate that UFM excels in generalization and performance across various feature matching tasks. The code will be released at: https://github.com/LiaoYun0x0/UFM.

## 1 Introduction

Feature matching, as a fundamental task in computer vision, serves to establish correspondences between local features in different images, facilitating the advancement of various downstream applications such as image fusion [1,2], image stitching [3,4], and 3D reconstruction [5,6], among others. With the proliferation of computer vision applications, the demand for precise feature matching of diverse multi-modal images has grown significantly. When researchers address the task of specific modal images, they are required to identify the appropriate feature matching method tailored to the corresponding modal image. This process typically involves utilizing substantial amounts of relevant training data, leading to substantial resource consumption. Thus, the development of a unified pre-trained comprehensive model for feature matching across a wide range of modals becomes increasingly imperative.

The concept of pretraining-fine-tuning methods initially emerged in the realm of natural language processing [7,8]. Given its remarkable performance, this approach was swiftly

https://github.com/hpatches,
http://www.ok.sc.e.titech.ac.jp/INLOC/,
https://data.ciirc.cvut.cz/public/projects/
2020VisualLocalization/Aachen-Day-Night/,
https://mediatum.ub.tum.de/1474000,
https://cs.nyu.edu/silberman/datasets/
nyudepthv2.html, http://matthewalunbrown.
com/nirscene/nirscene.html, https://github.com/
AmberHen/WHU-OPT-SAR-dataset,
http://www.ti.uni-bielefeld.de/html/people/
ddiffert/databasesuvg.html,
https://brainweb.bic.mni.mcgill.ca/brainweb/.

**Funding:** This work was supported by the National Natural Science Foundation of China under Grant 61976124 and 62372077 to M.L. This work was also supported by the Scientific Research Fund of Yunnan Provincial Education Department under Grant 2021J0007 to Y.L. and Q.D.

**Competing interests:** The authors have declared that no competing interests exist.

extended to the field of computer vision [9,10] and multimodal applications [11,12], offering a means to achieve more efficient results with reduced resources and time. Nevertheless, within the domain of multi-modal image feature matching, a comprehensive and effective unified framework has been notably absent, limiting the ability of researchers to make rapid advancements in multi-modal image feature matching through fine-tuning on a pre-trained foundational model.

Feature matching tasks can be broadly categorized into two types: matching features within images of the same modal and matching features across images of different modals. As illustrated in Fig 1, feature matching tasks for images of the same modal often involve significant variations in sensor positions and angles. For instance, the left image might capture the front of a building while the right image could be taken from the left side, exhibiting notable disparities in time, location, and angle between the two images. The primary objective in matching features within images of the same modal is to precisely align the poses of objects depicted in different pictures. Conversely, in the context of feature matching tasks involving images of different modals, the sensor positions typically exhibit minimal differences, serving to leverage the complementary information provided by the distinct modals present in the images. Previous studies have introduced numerous techniques [13–15] for facilitating feature matching within images of the same modal, along with several methods [16,17] dedicated to achieving feature matching across specific pairs of modals. While some approaches [18,19] have attempted to address feature matching across multimodal images [20], their applicability is often restricted to specific image modals, necessitating extensive training for specialized tasks.

In this paper, we introduce the Unified Pre-trained Feature Matching (UFM) model, designed to facilitate multi-modal image feature matching across a wide range of image modals. UFM not only enables feature matching within images of the same modal but also supports feature matching across various modals. This capability is achieved through the utilization of a Multi-Modal Image Assistant (MIA) Transformer. By incorporating a diverse set of modal assistants to aid the feedforward network within the standard transformer, the MIA Transformer can effectively capture specific modal information. Furthermore, the model

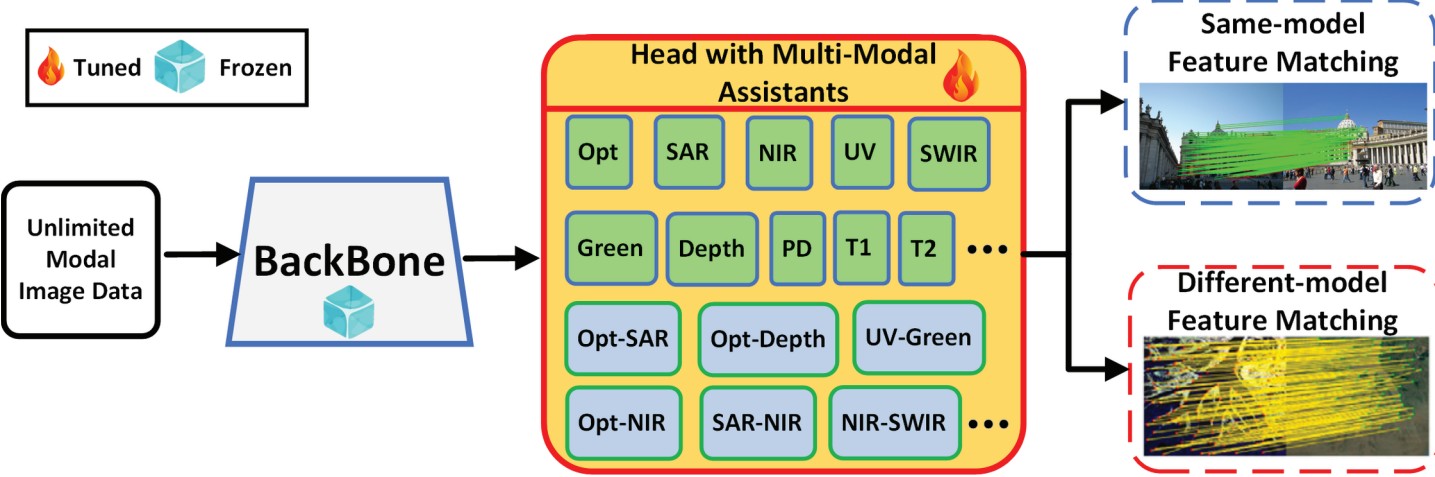

**Fig 1. Illustration of feature matching of UFM.** When working with specific data, the pre-trained backbone is frozen, and only the corresponding modal assistants need to be fine-tuned for feature matching. The multi-modal assistants contain both same-modal assistants and different-modal matching assistants.

leverages cross-modal shared self-attention and cross-attention mechanisms to capture inter-modal information. The modal assistants encompass two distinct types: those designed for the same image modal and those tailored for different image modals. Leveraging its modeling flexibility, the MIA Transformer, equipped with shared parameters, can be repurposed for various tasks.

To address the disparities in data distribution among different modal image data and to compensate for the scarcity of specific modal image data, we have devised a data augmentation methodology to expand the datasets of various modal images. Additionally, UFM incorporates a staged pre-training strategy, commencing with an initial pre-training phase on the same modal data and followed by subsequent pre-training on a dataset inclusive of various modals. Given that the feature matching data for images of the same modal significantly outweighs the corresponding data for different modals, this staged pre-training strategy significantly augments the data available for pre-training, thereby enhancing the generalization capabilities of UFM. Experimental results attest to the exceptional performance of UFM in both the feature matching tasks of images from the same modal and the feature matching tasks of images from different modals.

Our main contributions are as follows:

1. We introduce a unified pre-trained model for multi-modal image feature matching, denoted as UFM, which is adaptable to a wide array of modal image feature matching tasks.
2. We present the Multi-Modal Image Assistant (MIA) Transformer, a novel framework that captures specific modal information through modal assistants. Furthermore, MIA Transformer enhances the efficacy of feature matching for different modal images by facilitating cross-modal feature fusion.
3. Our proposed method demonstrates the ability to handle feature matching tasks for images of the same modal with significant disparities in sensor poses, as well as for feature matching tasks between images of different modals.
4. Leveraging data augmentation techniques, we expand the datasets of modal images with limited data, while the staged pre-training strategy significantly enhances the effectiveness of pre-training for multi-modal image feature matching.

## 2 Related work

In the realm of feature matching, methods can be categorized into two types: detector-based and detector-free. Beyond methodological distinctions, tasks in this domain can also be classified into two broad categories: conventional feature matching tasks and multi-modal image feature matching tasks.

**Detector-based feature matching.** The detector-based method unfolds in three key stages: feature detection, feature description, and feature matching. In the feature detection and description stage, interest points with descriptors are generated, followed by the establishment of point-to-point correspondences through a suitable matching algorithm. Detector-based methods can be further categorized into handcrafted descriptor methods and deep learning descriptor methods. Among the prominent handcrafted descriptor methods is SIFT [21], renowned for its simplicity and efficiency, making it versatile across various computer vision tasks. Subsequent works [22–24] have iteratively improved upon the SIFT algorithm, enhancing its performance. Deep learning descriptor methods leverage convolutional neural networks to glean deep features and capture nonlinear expressions, uncovering valuable hidden information. Sarlin et al. [25] introduced SuperGlue, featuring a flexible attention-based context aggregation mechanism capable of jointly reasoning about the underlying 3D

scene and feature assignment. Jiang et al. [26] proposed GLMNet, a graph learning-matching network utilizing graph convolutional networks to adaptively learn optimal graphs for the graph matching task.

**Detector-free feature matching.** The detector-free method eliminates the need for extracting interest points, opting instead to directly extract features in image pairs through transformers. This category can be further divided into semi-dense matching methods and dense matching methods. Semi-dense matching methods typically employ a coarse-to-fine matching process, achieving final precise matching results on 1/2 size feature maps. Notably, a pioneering contribution is the LoFTR algorithm by Shen et al. [27], which introduces global receptive fields in Transformers to produce semi-dense matches, particularly excelling in low-texture regions. Subsequent algorithms [28–32] have iteratively built upon and improved LoFTR as a baseline. Dense matching methods, on the other hand, extract all matches between views directly on the original image, aiming to estimate each matched pair of pixels. Truong et al. introduced PDC-Net [33] and PDC-Net+ [34], formulating bias estimates in a probabilistic manner and pairing proposed feature correspondences with deterministic estimates via a mixture model. Edstedt et al. presented DKM [35] and RoMa [36]. While DKM achieves superior matching accuracy by utilizing depthwise separable large kernels and local correlations with stacked feature maps as input, RoMa combines the strengths of semi-dense and dense matching methods. It designs a coarse-to-fine dense matching framework and achieves state-of-the-art matching accuracy. Despite the higher accuracy of dense matching methods compared to semi-dense counterparts, they come at the cost of increased computational resources and training time. Considering these factors comprehensively, we adopt a semi-dense matching framework in designing UFM.

**Multi-modal image feature matching.** The significant disparities among sensors in multi-modal images pose challenges for conventional matching methods, rendering them less applicable for multi-modal image feature matching. Consequently, numerous scholars have introduced algorithms specifically tailored for this purpose. Zhu et al. [37] presented R2FD2, a detector-based multi-modal image feature matching algorithm. Initially, they employed the reproducible feature detector MALG to identify interest points and subsequently utilized the feature descriptor RMLG for feature representation and matching. Hu et al. [19] proposed a multi-scale structural feature transform (MSFT) method designed for multi-modal image matching. This approach detects scale-invariant feature points based on the Gaussian difference image pyramid of the phase congruency map, addressing challenges arising from nonlinear radiation distortion. In the detector-free domain, Di et al. [18] introduced FeMIP, a multi-modal feature matching algorithm. Employing a coarse-to-fine approach, FeMIP achieves accurate feature matching. Notably, it incorporates a policy gradient method to address issues related to the discreteness of matching. While there exist several commendable multi-modal image feature matching algorithms, they often exhibit limitations in terms of the types of multi-modal images they are suited for. Furthermore, these algorithms may require substantial training on specific datasets and may not be well-suited for addressing feature matching challenges within images of the same modal. Therefore, the development of a unified feature matching algorithm, accommodating the majority of modal images, is of utmost importance.

**Pre-training – fine-tuning.** As the transformer paradigm has evolved, the pretraining-fine-tuning approach has become instrumental in advancing the state of the art across diverse domains such as natural language processing [38,39], computer vision [40,41], and multi-modal tasks [42,43]. Zaken et al. [44] introduced BitFit, a sparse fine-tuning method tailored for natural language processing. Leveraging the modification of bias terms exclusively, this method demonstrates remarkable performance on extensive training data. Sohn et al. [45]

presented a method for learning vision transformers through generative knowledge transfer. Their approach incorporates a novel prompt design that strategically places learnable tokens within image token sequences. Radford et al. [11] proposed CLIP, a multi-modal pretraining-fine-tuning model that exhibits effective knowledge transfer across tasks without the need for dataset-specific training. Despite the notable success of the pre-training-fine-tuning technique in various domains, its application in the realm of multi-modal image feature matching has been limited. Consequently, we have devised a unified large model, UFM, for multi-modal image feature matching, achieving comprehensive pre-training on extensive multi-modal image data.

## 3 Methodology

Given an image pair from most modals, the Unified Feature Matching (UFM) approach can effectively derive its image pair representation using a Multimodal Image Assistant (MIA) Transformer network. The UFM method exhibits remarkable versatility, enabling it to address feature matching tasks not only within the same modal but also across different modals, even when the sensor positions vary significantly. Additionally, in handling the feature matching tasks of diverse modal images, the process of cross-modal feature fusion can further enhance the overall effectiveness of the approach.

### 3.1 Multi-modal image assistants transformer

Similar to previous methods [18,27,28], UFM adopts a coarse-to-fine dense matching approach for feature matching. The input image pairs undergo processing via the FPN network to generate features at 1/8th and 1/2th sizes. The 1/8-size features are trained using the augmented GT matrix as labels, yielding coarse matching results at the patch level. Subsequently, the 1/2 size features are precisely matched with the coarse matching results to derive the final dense matching results at the pixel level. Distinguishing itself from other methods, UFM introduces a novel architecture, departing from the conventional use of transformers in coarse and fine matching stages.

In this study, we propose a novel Multi-modal Image Assistant (MIA) Transformer tailored for feature matching tasks, as depicted in Fig 2. The UFM model encompasses all the processes involved in image enhancement, feature extraction, coarse matching, and fine matching. The MIA Transformer is utilized in both the coarse and fine matching stages. The MIA Transformer integrates a multi-modal image assistant with a generic feedforward network. The Unified Feature Matching (UFM) framework comprises a total of L layers. Given the output vector G_(l-1) from the generic feedforward network of the preceding layer and the output vector A_(l-1) from the assistant feedforward network of the previous layer, the approach leverages multi-head self-attention and cross-attention (MSCA) mechanisms, shared across modals, to align the content of a pair of images. The input vector $V'_l$ for each layer can be computed as follows:

$$V'_l = \text{MSCA}\left(LN\left(G_{l-1} + A_{l-1}\right)\right) + G_{l-1} + A_{l-1}, \tag{1}$$

where LN is short for layer normalization. MIA_FFN is designed to select the appropriate assistant from a collection of multiple modal assistants for processing the input vector. Notably, it encompasses two distinct types of modal assistants: those tailored for the same modal and those dedicated to different modals. When the input consists of image vectors from the same modal, the corresponding assistant associated with that modal is employed

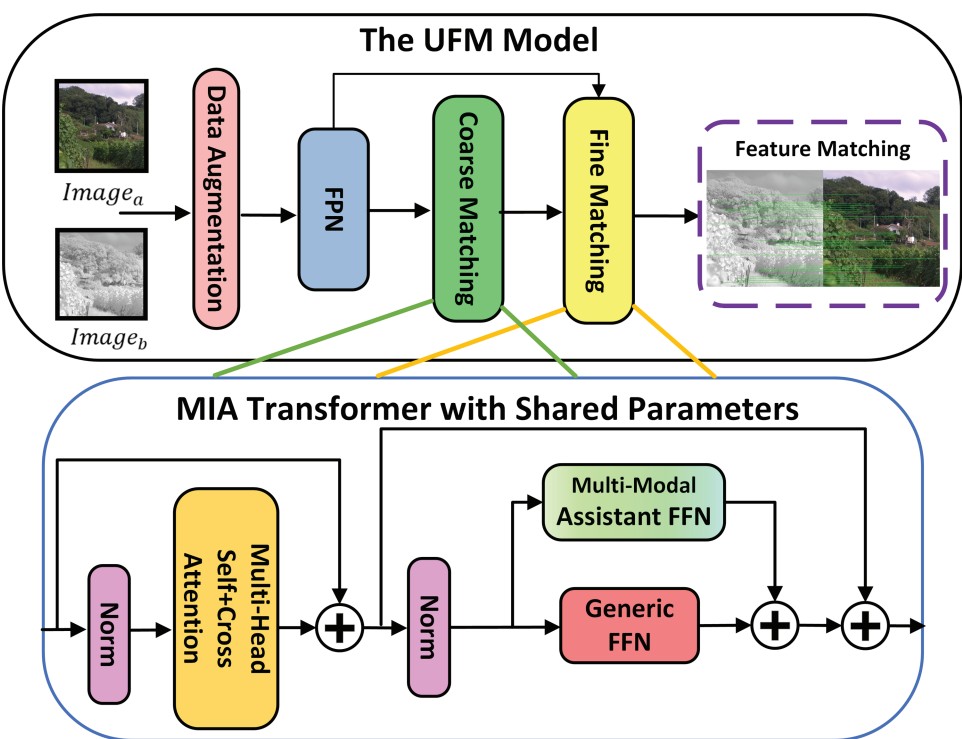

**Fig 2. The overview of UFM and the MIA transformer.** The UFM model encompasses all the processes involved in image enhancement, feature extraction, coarse matching, and fine matching. The MIA Transformer is utilized in both the coarse and fine matching stages.

for encoding the images. Conversely, in cases where the input includes image vectors from diverse modals, such as optical-SAR, the optical and SAR assistants are utilized to encode the respective modal vectors at the underlying Transformer layer. Subsequently, the optical-SAR assistant is employed at the top layer to capture more comprehensive modal interactions. The output vector V_l for each layer can be computed as follows:

$$V_l = MIA\_FFN\left(LN\left(V'_l\right)\right) + V'_l. \tag{2}$$

## 3.2 Data augmentation

UFM necessitates a comprehensive pre-training process involving images from all modals. However, significant discrepancies often exist in the availability of image data across different modals. For instance, optical images commonly offer a wealth of data, whereas long-infrared images may be relatively scarce. To address the nonuniformity in data distribution among different modals and to mitigate the scarcity of data in certain modals, this paper introduces a data augmentation technique. The primary objective of this technique is to achieve a more balanced data distribution for the image dataset of each modal and enhance the generalizability of UFM.

As shown in Fig 3, given an image pair $Image_a$ and $Image_b$, a sequence of data augmentation procedures is initially applied. Subsequently, these enriched image pairs are utilized to produce a comprehensive pixel matching label GT_matrix. The augmentation process includes mirroring, flipping, and rotating the input images, significantly enhancing the

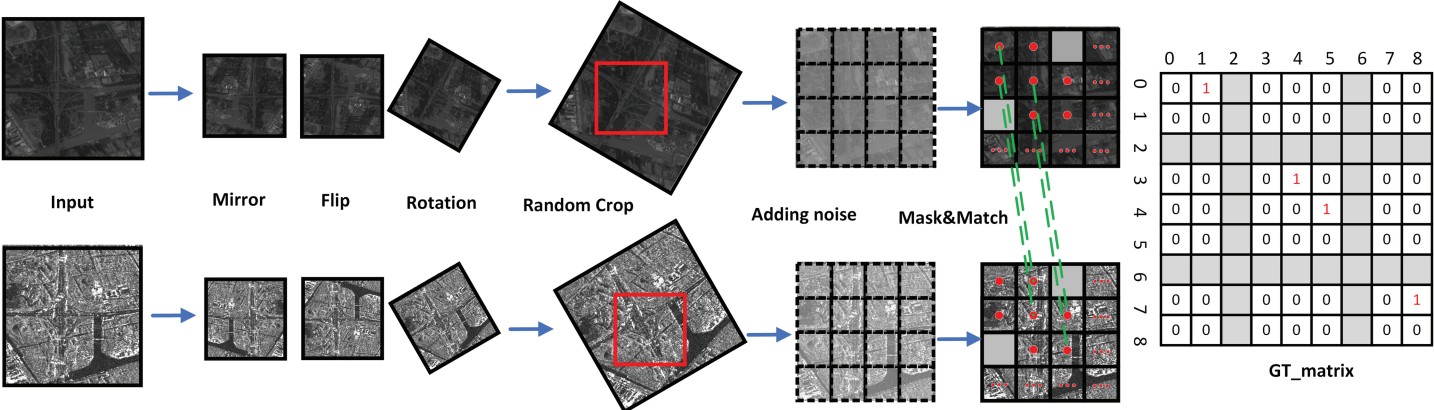

**Fig 3. Data augmentation is applied both geometrically and in terms of intensity.** Geometrically, the images are mirrored, flipped, rotated, and randomly cropped. For intensity augmentation, random noise is added, and random masking is applied. Finally, a square matrix (GT matrix) is used to represent the correspondence of matching points between the two images. The GT_matrix is a square matrix of $N \times N$ dimensions. $GT(i, j)$ represents the element of the $i$th row and the $j$th column in the GT matrix. The shown input image pairs take optical and SAR image pairs as an example.

diversity within the dataset. In addition, the processed image is randomly cropped. Finally, random noise is added to the cropped block map, and some pixels are randomly masked [20].

The cropped pair of images are defined as $I_a$ and $I_b$. The sizes of $I_a$ and $I_b$ are the same, and both their height and width are denoted by h and w, respectively. The image is divided into N image patches. The patch coordinates can be defined as $(i_a^p, j_a^p)$, where $i_a^p = 0, \dots, w/p - 1$; $j_a^p = 0, \dots, h/p - 1$ and p is the size of the patch. We applied a random mask to these image patches and the proportion of them is between 20% and 40%. The location of the random mask is defined as M and can be expressed as

$$\mathcal{M} \leftarrow Rand(i_a^p, j_a^p), 0.2N \leq |\mathcal{M}| \leq 0.4N, \qquad (3)$$

For positions that are not masked, the central points of the patches in $I_a$ can be defined as $(i_a^c, j_a^c)$, which can be calculated as

$$(i_a^c, j_a^c) = \begin{cases} i_a^c = i_a^p \times p + \frac{p}{2} \\ j_a^c = j_a^p \times p + \frac{p}{2}, \end{cases} \qquad (4)$$

The coordinates of the points in $Image_b$ corresponding to the $(i_a^c, j_a^c)$ are defined as $(i_b^c, j_b^c)$. They can be obtained by running a series of data augmentation operations (Mirror, flip, rotate, etc.) in reverse for coordinates $(i_a^c, j_a^c)$. Then the corresponding patch coordinates of them can also be extracted as

$$\left(i_b^p, j_b^p\right) = \begin{cases} i_b^p = \left[\frac{i_b^c + 1}{p}\right] \\ j_b^p = \left[\frac{j_b^c + 1}{p}\right], \end{cases} \qquad (5)$$

where [o] means round down o.

Then we define a GT matrix to represent the matching of the image patches after data augmentation. $I_a$ partially overlaps with $I_b$, so the corresponding patch coordinates $(i_b^p, j_b^p)$ may be inside or outside of $I_b$. If $(i_b^p, j_b^p)$ is in image $I_b$, then

$$\text{GT}\left(j_a^p \times \frac{w}{p} + i_a^p, j_b^p \times \frac{w}{p} + i_b^p\right) = 1. \tag{6}$$

where $\text{GT}(i, j) = 1$ indicates that the unmasked $i$th patch in $I_a$ matches the unmasked $j$th patch in $I_b$.

### 3.3 Pre-training

Presented in Fig 4, our proposed staged pre-training strategy aims to enhance the image matching model's performance by leveraging a large-scale image dataset from the same modality. The pre-training is divided into 3 stages: (1) pre-train the general FFN, (2) pre-train all X-X assistants, and (3) pre-train all X-Y assistants. In stage 1, given that optical images provide a rich source of feature matching data, we initially conduct pre-training on multi-head attention and the generic feedforward network (FFN) using a substantial collection of pure optical images. In stage 2, We pre-train feature matching for all of the same modal images (X-X). The pre-training of all modal images (X-X, Y-Y, Z-Z⋯) in the stage 2 can be performed in parallel. We freeze multi-head self+cross attention at this stage, which greatly improves the efficiency of training. In stage 3, we further pre-train the feature matching of all cross-modal images (X-Y) on the basis of stage 2. At this stage, we adjusted all the attention and corresponding FFNS to maximize cross-modal matching. The three FFNS (X-X, Y-Y, X-Y) corresponding to the two modal images are pre-trained simultaneously.

A notable consideration is the adoption of the concept of frozen attention blocks, as previously introduced in [46,47], to potentially enhance the pre-training for the same modal. Consequently, during the pre-training phase for the same modal, we retain the parameters of the multi-head attention and other modal assistants, exclusively training the assistant FFN tailored to the specific modal under consideration. In the process of training feature matching assistants for different modals, it is essential to conduct separate pre-training for each modal, ensuring that the parameters are fine-tuned accordingly. Notably, in scenarios involving different modals, such as Opt-SAR, the parameters for multi-head attention are not frozen, facilitating a more comprehensive adaptation to the diverse modals.

Conventional general multi-modal image matching methods are typically trained solely on limited image data from various modals, posing a challenge in achieving optimal results.

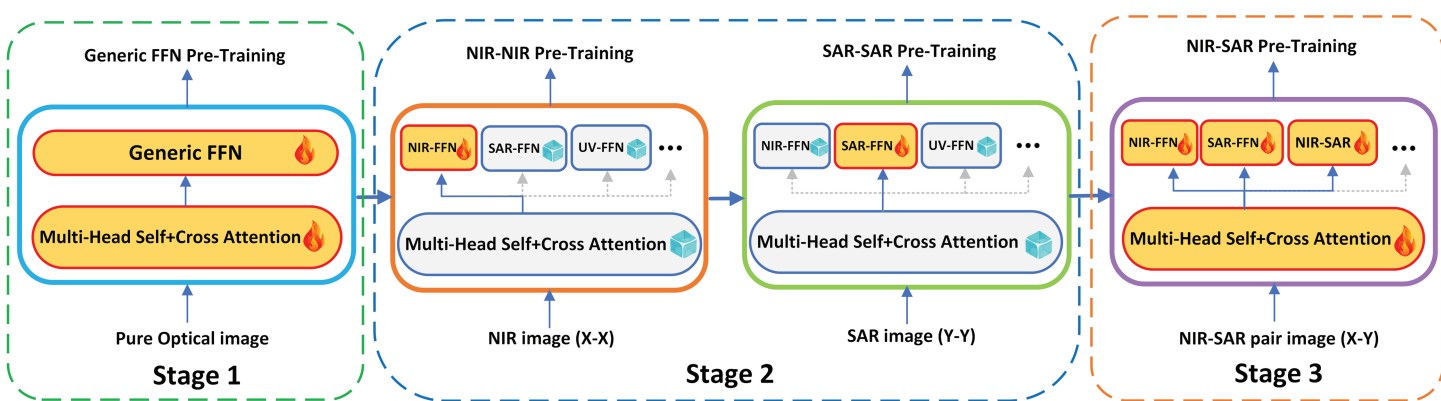

**Fig 4. Illustration of pre-training.** The pre-training of NIR and SAR images is taken here as an example. Pre-training consists of 3 stages: (1) pre-train the general FFN, (2) pre-train all X-X assistants, and (3) pre-train all X-Y assistants.

In contrast, datasets containing images of the same modal are comparatively more accessible. Leveraging the staged pre-training strategy enables the model to undergo initial pre-training on a same-modal dataset, followed by subsequent pre-training on datasets encompassing different modals. This approach significantly augments the volume of data available for pre-training and substantially enhances the model's overall generalization capacity.

By incorporating this staged pre-training strategy, the model can effectively leverage the advantages of the larger same-modal datasets during the initial training phase. Subsequently, through continued pre-training on diverse-modal datasets, the model can gain a more comprehensive understanding of the variations and nuances across different modals, thus improving its adaptability and performance across a broader range of modals.

In the specific pre-training process of feature matching, UFM is also trained in two stages: coarse matching and fine matching. In coarse matching, dual-softmax is used for training, which can be extracted as

$$\text{Loss}_c = -\frac{1}{n} \sum_k \left[ GT_{i,j} \cdot \log(P(i,j)) + \left(1 - GT_{i,j}\right) \cdot \log(1 - P(i,j)) \right] \qquad (7)$$

where $GT_{i,j}$ denotes the GT matrix, $P(i,j)$ represents the probability of the correct matching and n is the number of feature points.

Following [49], we used the epipolar loss. The epipolar constraint states that $j^T F_i = 0$ holds if i and j are truly matched, where $F_i$ can be interpreted as the epipolar line corresponding to i in $I_b$. The epipolar loss is defined as the distance between the predicted corresponding position and the ground-truth epipolar line:

$$L_{ep}(i) = \text{dist}\left(h_{1\to2}(i), F_i\right), \qquad (8)$$

where $h_{1\to2}(i)$ is the predicted correspondence in $I_b$ for the point i in $I_b$, and $\text{dist}(\cdot, \cdot)$ is the distance between a point and a line.

The epipolar loss itself only encourages the predicted match to lie on the epipolar line rather than close to the ground truth correspondence. We also need to introduce a cycle consistency loss to encourage the forward and backward mapping of a point to be spatially close to itself:

$$L_{cy}(i) = \left\| h_{2\to1}\left(h_{1\to2}(i)\right) - i \right\|_2. \qquad (9)$$

For a pair of enhanced images $I_a$ and $I_b$, the dense feature descriptors extracted by MIA transformer are defined as $M_1$ and $M_2$. To compute the correspondence for a query point i in $I_a$, we correlate the feature descriptor at i, denoted by $M_1(i)$, with all of $M_2$. A 2D pixel location distribution of $I_b$ is obtained, indicating the probability corresponding to each location and i in $I_a$. The probability distribution can be expressed as $p\left(x \mid i, M_1, M_2\right)$. A single 2D match can then be computed as the expectation of this distribution:

$$\hat{j} = h_{1\to2}(i) = \sum_{x\in I_b} x \cdot \frac{\exp\left(M_1(i)^T M_2(x)\right)}{\sum_{y\in I_b} \exp\left(M_1(i)^T M_2(y)\right)} \qquad (10)$$

where y varies over the pixel grid of $I_b$.

The loss at each point is re-weighted using the total variance $\sigma^2(i)$ as an uncertainty measure. The loss function for fine matching is the weighted sum of the epipolar and cycle consistency loss of the n sampled query points:

$$\text{Loss}_f = \sum_{m=1}^{n} \frac{1}{\sigma^2(i^m)} \left[ L_{ep}(i^m) + \lambda L_{cy}(i^m) \right].$$ (11)

The overall loss function Loss is composed of $\text{Loss}_c$ and $\text{Loss}_f$, which can be expressed as

$$Loss = \alpha \, \text{Loss}_c + \beta \, \text{Loss}_f.$$ (12)

### 3.4 Fine-tuning on different feature matching tasks

After comprehensive pre-training, UFM only requires fine-tuning on the corresponding dataset to achieve excellent matching results. For instance, when dealing with a brand new multimodal dataset, other methods typically need to be pre-trained on the entire training set. In contrast, UFM usually requires only about 1/10 of the training data for fine-tuning to deliver superior matching performance. This fine-tuning process includes both within-modal feature matching and cross-modal image feature matching. This critical stage ensures that the model effectively adapts to the unique characteristics and complexities of the matching task.

**Fine-tuning of feature matching for the same modal.** As illustrated in Fig 5, UFM facilitates the fine-tuning of image feature matching for individual modals. During the fine-tuning process for the same modal, UFM selectively freezes the parameters associated with the generic feedforward network (FFN) and multi-head attention, focusing solely on refining

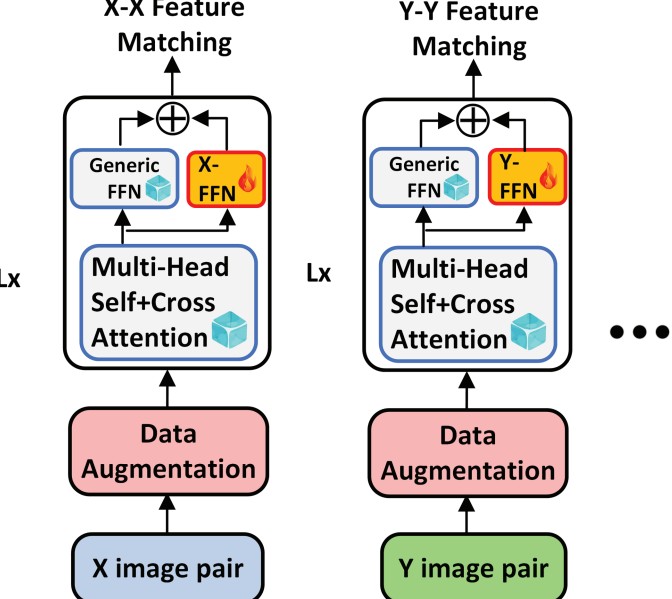

**Fig 5. Fine-tuning on same-model feature matching tasks.** The X-FFN and Y-FFN represent the assistants of any two kinds of pre-trained different modal images in the second stage of Fig 4. The fine-tuning of the X-modal image and the fine-tuning of the Y-modal image are independent of each other.

the assistant relevant to the corresponding modal. This modal assistant works in conjunction with the general FFN, utilizing a residual structure. By employing this strategy, the model can capitalize on the robust capabilities of the general FFN while making specific adjustments tailored to the data characteristics of the current modal. This approach not only enhances the model's adaptability and performance within the same modal but also significantly reduces the computational resources required for training, ensuring a more efficient and effective fine-tuning process. **Fine-tuning of feature matching across different modals.** Illustrated in Fig 6, the fine-tuning process for image feature matching across different modals in UFM involves two distinct stages. In the first stage, the assistant FFNs from two modals aid the generic FFN in extracting features. In the second stage, the assistant FFNs for modal matching assist the general FFN in executing feature matching. Throughout both stages, the parameters of the multi-head attention and general FFN remain frozen, and only the modal-specific assistant is subject to fine-tuning. Specifically, the first stage encompasses L-M layers, while the second stage comprises M layers. The total number of layers corresponds to that used for fine-tuning image feature matching within the same modal. This two-stage strategy not only leverages the specialized modal assistant to facilitate feature extraction but also enhances the effect of feature matching across different modals through cross-modal feature fusion. By integrating these complementary stages, UFM effectively optimizes the feature matching process, ensuring improved performance and enhanced adaptability across diverse modals.

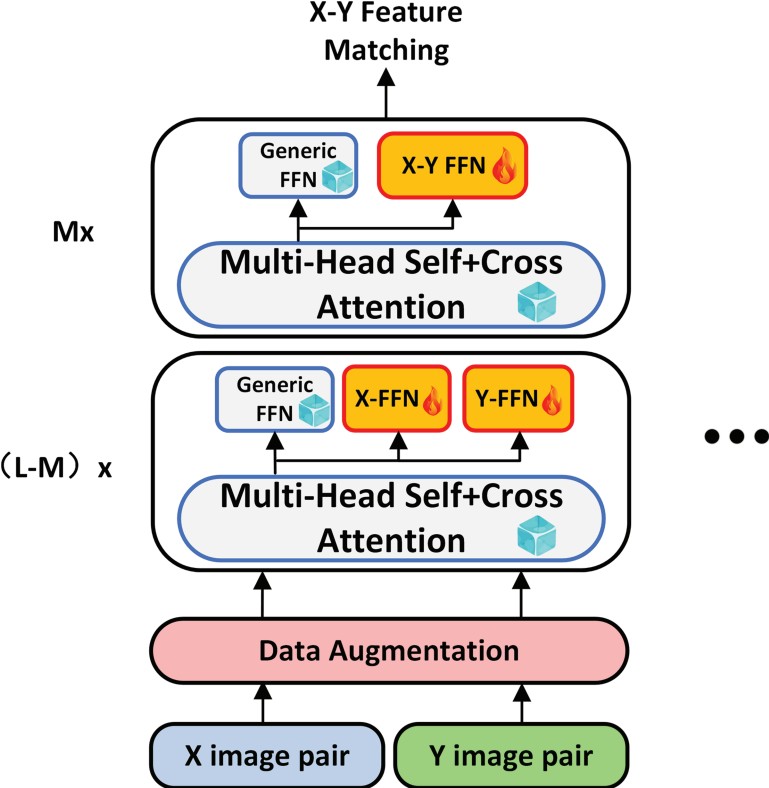

**Fig 6. Fine-tuning on different-model feature matching tasks.** The X-FFN and Y-FFN represent the assistants of any two kinds of pre-trained different modal images in the second stage of Fig 4. The X-Y FFN represent the assistant of the pre-trained different modal images in the third stage of Fig 4.

## 4 Experiments

We pre-train the UFM on large-scale multimodal image data and evaluate the models qualitatively and quantitatively on different feature matching tasks.

### 4.1 Pre-training setup

Our pre-training data mainly consists of these datasets: MegaDepth [50], ScanNet [51], YFCC100M [52], OSCD [53], MRSI [54], MRSIs [37], DIRSIG [55], MS-SAR LCZ [56], Retina [57], BrainWeb [58], VIS–NIR [59], WxBS [60] and image–paint [61]. There are about 2 million image pairs in the pre-training data. After pre-training, we only need to use about 1/10 of the training set of the corresponding dataset for fine-tuning.

The model consists of 9 layers transformers with 768 hidden size. The model is trained by AdamW optimizer with $\alpha = 0.9$, $\beta = 0.98$. The learning rate is $1 \times 10^{-4}$. The pre-training of multi-modal image feature matching takes about a week using 8 Nvdia Tesla V100 32GB GPU cards.

### 4.2 Estimation on same-modal images feature matching

In the feature matching evaluation of images from the same modal, we mainly verify the feature matching between images with large camera pose differences. As shown in Fig 7, We compare the feature matching of different methods on Hpatches [62], InLoc [63] and Aachen Day-Night v1.1 [64] datasets. The InLoc and Aachen Day-Night v1.1 datasets are just the datasets for testing and do not have any training data. We fine-tuned on the Hpatches dataset using 1/10 of the training data.

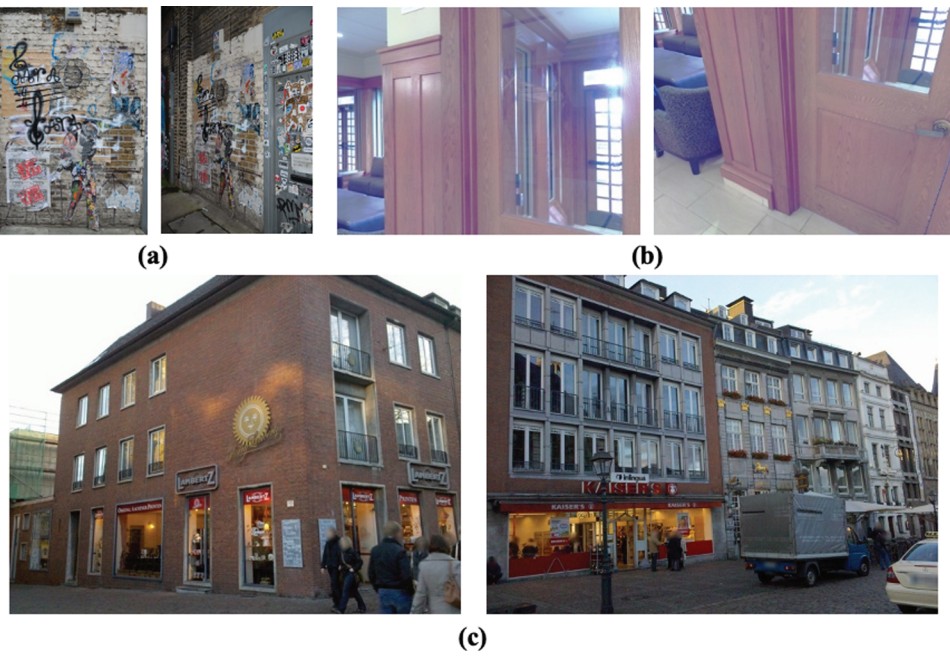

**Fig 7. Examples of the image pairs of the same modal.** (a) The Hpatches dataset. (b) The InLoc dataset. (c) The Aachen Day-Night v1.1 dataset.

**Table 1. Evaluation on HPatches for homography estimation.**

| Method | Overall | Illumination | Viewpoint | Matches |
|---|---|---|---|---|
| | Accuracy (%, $\epsilon < 1/3/5px$) | | | |
| SP [48] | 0.46/0.78/0.85 | 0.57/0.92/0.97 | 0.35/0.65/0.74 | 1.1K |
| D2Net [71] | 0.38/0.71/0.82 | 0.66/0.95/0.98 | 0.12/0.49/0.67 | 2.5K |
| R2D2 [72] | 0.47/0.77/0.82 | 0.63/0.93/0.98 | 0.32/0.64/0.70 | 1.6K |
| ASLFeat [73] | 0.48/0.81/0.88 | 0.62/0.94/0.98 | 0.34/0.69/0.78 | 2.0K |
| SP [48] + SuperGlue [25] | 0.51/0.82/**0.89** | 0.60/0.92/0.98 | **0.42**/0.71/0.81 | 0.5K |
| SP [48] + CAPS [49] | 0.49/0.79/0.86 | 0.62/0.93/0.98 | 0.36/0.65/0.75 | 1.1K |
| SIFT + CAPS [49] | 0.36/0.77/0.85 | 0.48/0.89/0.95 | 0.26/0.65/0.76 | 1.5K |
| SparseNCNet [71] | 0.36/0.65/0.76 | 0.62/0.92/0.97 | 0.13/0.40/0.58 | 2.0K |
| Patch2Pix [74] | 0.50/0.79/0.87 | 0.71/0.95/0.98 | 0.30/0.64/0.76 | 1.3K |
| LoFTR [27] | 0.55/0.81/0.86 | 0.74/0.95/0.98 | 0.38/0.69/0.76 | 4.7K |
| MatchFormer [28] | 0.55/0.81/0.87 | 0.75/0.95/0.98 | 0.37/0.68/0.78 | **4.8k** |
| UFM | **0.56/0.82**/0.87 | **0.83/0.99/0.99** | 0.37/**0.69/0.78** | **4.8k** |

**Table 2. Visual localization evaluation on the Aachen Day-Night v1.1.**

| Method | Day | Night |
|---|---|---|
| | $(0.25m, 2°) / (0.5m, 5°) / (1.0m, 10°)$ | |
| PixLoc [75] | 61.7 / 67.6 / 74.8 | 46.9 / 53.1 / 64.3 |
| HSCNet [76] | 71.1 / 81.9 / 91.7 | 40.8 / 56.1 / 76.5 |
| HFNet [77] | 76.2 / 85.4 / 91.9 | 62.2 / 73.5 / 81.6 |
| Patch2Pix [74] | 86.4 / 93.0 / 97.5 | 72.3 / 88.5 / 97.9 |
| ISRF [78] | 87.1 / 94.7 / 98.3 | 74.3 / 86.9 / 97.4 |
| RLOCS [79] | 88.8 / 95.4 / 99.0 | 74.3 / 90.6 / 98.4 |
| SP [48] + SuperGlue [25] | 89.9 / 96.1 / **99.4** | 77.0 / **90.6** / **100.0** |
| LoFTR [27] | 88.7 / 95.6 / 99.0 | 78.5 / **90.6** / 99.0 |
| UFM | **90.1 / 96.2** / 99.1 | **78.6** / **90.6** / 99.5 |

As shown in Table 1, we evaluate the effect of UFM and contrast algorithms in homography estimation on the HPatches benchmark set and report the proportion of accurate predictions with average corner error distances less than 1/3/5 pixels. Bold numbers in the table indicate the best results under the current metric. Compared with these excellent algorithms, UFM has achieved the best results under most indicators. Especially under the Illumination indicator, UFM achieved extremely excellent results.

To verify the visual localization capability of UFM, we also estimated the 6-DOF pose of a given image with respect to the corresponding 3D scene model. We evaluated different approaches on long-term visual localization benchmarks [65]. The focus is on benchmarking indoor scene changes and day/night changes. As shown in Tables 2 and 3, UFM is very competitive on both Aachen Day-Night v1.1 and InLoc datasets. UFM surpasses the compared methods under most metrics.

## 4.3 Estimation on different-modals images feature matching

In the evaluation of feature matching between images of different modals, seven datasets were used for testing. As shown in Fig 8, the data sets used for the test are: SEN 12 MS dataset [66], RGB-NIR Scene dataset [59], WHU-OPT-SAR dataset [67], Optical-SAR dataset [68], Brain-Web dataset [58], NYU-Depth V2 dataset [69] and UV-Green dataset [70]. We fine-tuned

**Table 3. Visual localization evaluation on the InLoc benchmark.**

| Method | DUC1 $(0.25m, 10°) / (0.5m, 10°) / (1.0m, 10°)$ | DUC2 |
|---|---|---|
| ISRF [78] | 39.4 / 58.1 / 70.2 | 41.2 / 61.1 / 69.5 |
| R2D2 [72] | 41.4 / 60.1 / 73.7 | 47.3 / 67.2 / 73.3 |
| COTR [80] | 41.9 / 61.1 / 73.2 | 42.7 / 67.9 / 75.6 |
| Patch2Pix [74] | 44.4 / 66.7 / 78.3 | 49.6 / 64.9 / 72.5 |
| SP [48] + SuperGlue [25] | 49.0 / 68.7 / 80.8 | 53.4 / 77.1 / 82.4 |
| LoFTR [27] | 47.5 / 72.2 / 84.8 | 54.2 / 74.8 / 85.5 |
| UFM | **51.5 / 73.8** / 85.9 | **54.9 / 74.8 / 86.3** |

on these datasets using 1/10 of the training data. UFM can handle most multi-modal feature matching problems. This test experiment mainly covers optical images, SAR images, NIR images, SWIR images, depth images, UV images, green images and medical images.

**Multiple modals of the same scene.** In the SEN12MS dataset, multiple images correspond to different modals of the same scene. We perform MMA evaluation and Homography estimation on images from different modals of SEN12MS. Except UFM, the other algorithms are fully trained. The results of MMA are shown in Fig 9. In this experiment, the average matching accuracy of different methods is calculated for pixel values ranging from 1 to 10. The higher and more left MMA curve indicates the better feature matching performance of the proposed method. The results of the Homography estimation are shown in Table 4. We report the regions under the cumulative curve (AUC) where corner error reaches the 3, 5, and 10 pixel thresholds, respectively. The higher the result of the Homography estimation, the better the effect of its feature matching. Through the experimental results, it can be found that UFM outperforms the compared algorithms in the vast majority of cases, indicating that it is highly competitive in feature matching of different modal images of the same scene.

**Multiple modals of the different scenes.** When performing feature matching of multi-modal images, most of the time it is necessary to deal with the task of different scenes. We tested feature matching on multiple modal images of various scenes. Except UFM, the other algorithms are fully trained. The results of the MMA evaluation and the results of the Homography estimation are shown in Fig 10 and Table 5, respectively. UFM also outperforms other excellent algorithms in most cases when only fine-tuning is performed. These experiments can prove that UFM has good feature matching performance and generalization while saving a lot of computing resources.

## 4.4 Visualization of image feature matching

**The images of the same modal.** As illustrated in Fig 11, in order to observe the matching effect of different methods more intuitively, the matches with less than 1 pixel error are represented by lines. To save space, we only show the matching results of the four methods with the best results. Compared with other methods, UFM has more correct matching lines, which qualitatively proves that UFM algorithm has a good effect on image feature matching of the same modal.

**The images of different modals.** The results of multi-modal image feature matching are generated, and the results of the four methods with the best results are presented in Fig 12. The datasets from top to bottom are: SEN 12 MS dataset, RGB-NIR Scene dataset, WHU-OPT-SAR dataset, Optical-SAR dataset, BrainWeb dataset, NYU-Depth V2 dataset and UV-Green dataset. In order to clearly compare the matching effects of different methods, we rotate

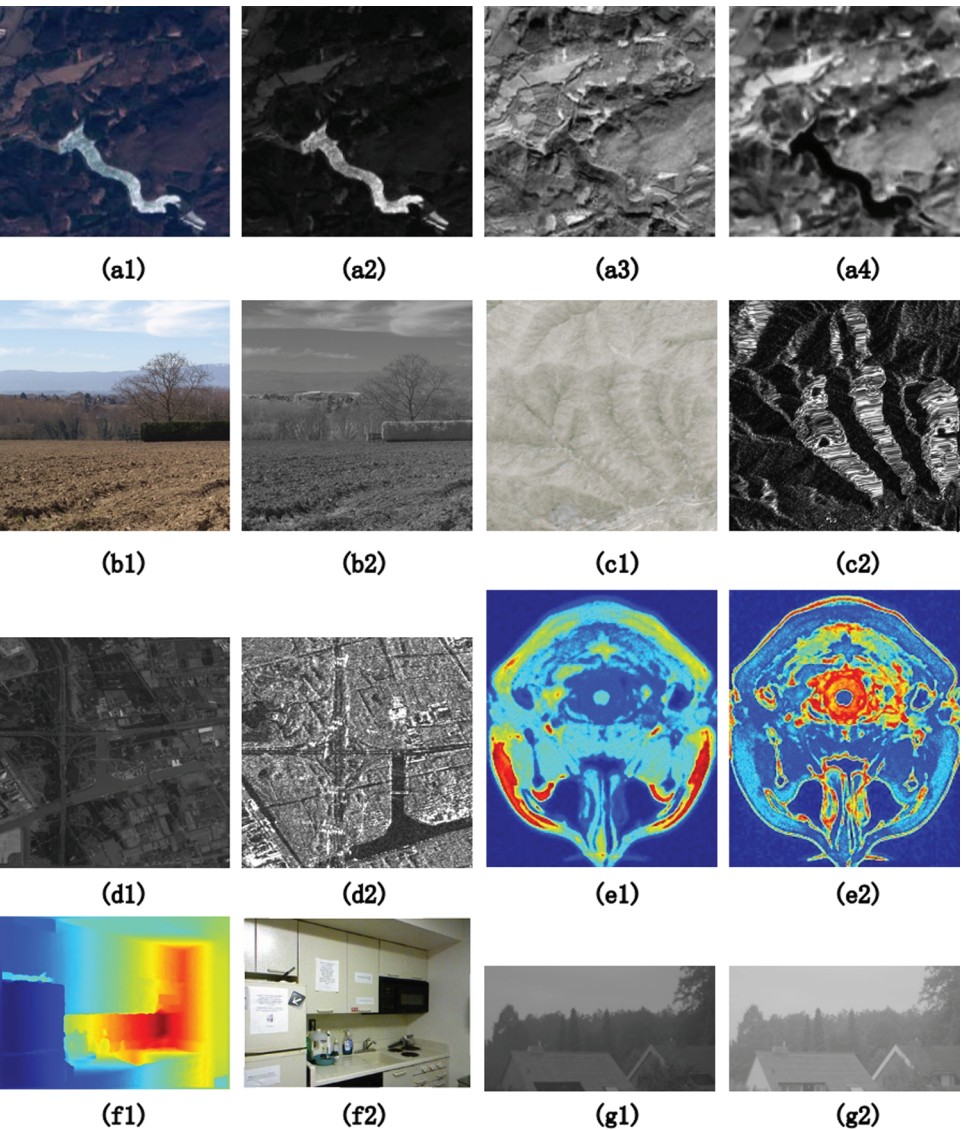

**Fig 8. Examples of the image pairs of different modals.** (a) The SEN 12 MS dataset. **a1.** Optical **a2.** SAR **a3.** NIR **a4.** SWIR (b) The RGB-NIR Scene dataset. **b1.** RGB **b2.** Near-Infrared (c) The WHU-OPT-SAR dataset. **c1.** Optical **c2.** SAR (d) The Optical-SAR dataset. **d1.** Optical **d2.** SAR (e) The BrainWeb dataset. **e1.** T1 **e2.** T2 (f) The NYU-Depth V2 dataset. **f1.** Depth **f2.** RGB (g) The UV/Green Image dataset. **g1.** UV **g2.** Green.

and crop the image data of many modals. The matches with less than 1 pixel error are represented by lines. Compared with other methods, the UFM algorithm obtains more correct matching connections, which qualitatively proves the competitiveness of the UFM algorithm in dealing with multi-modal image feature matching.

## 4.5 Estimation on the distance error test of feature matching

Although the lines of matched points with pixel error less than 1 are shown in Sect 4.4, it is not possible to show the specific error value for each pixel.To further objectively evaluate the matching accuracy of different methods, we calculated the average distance error of matching

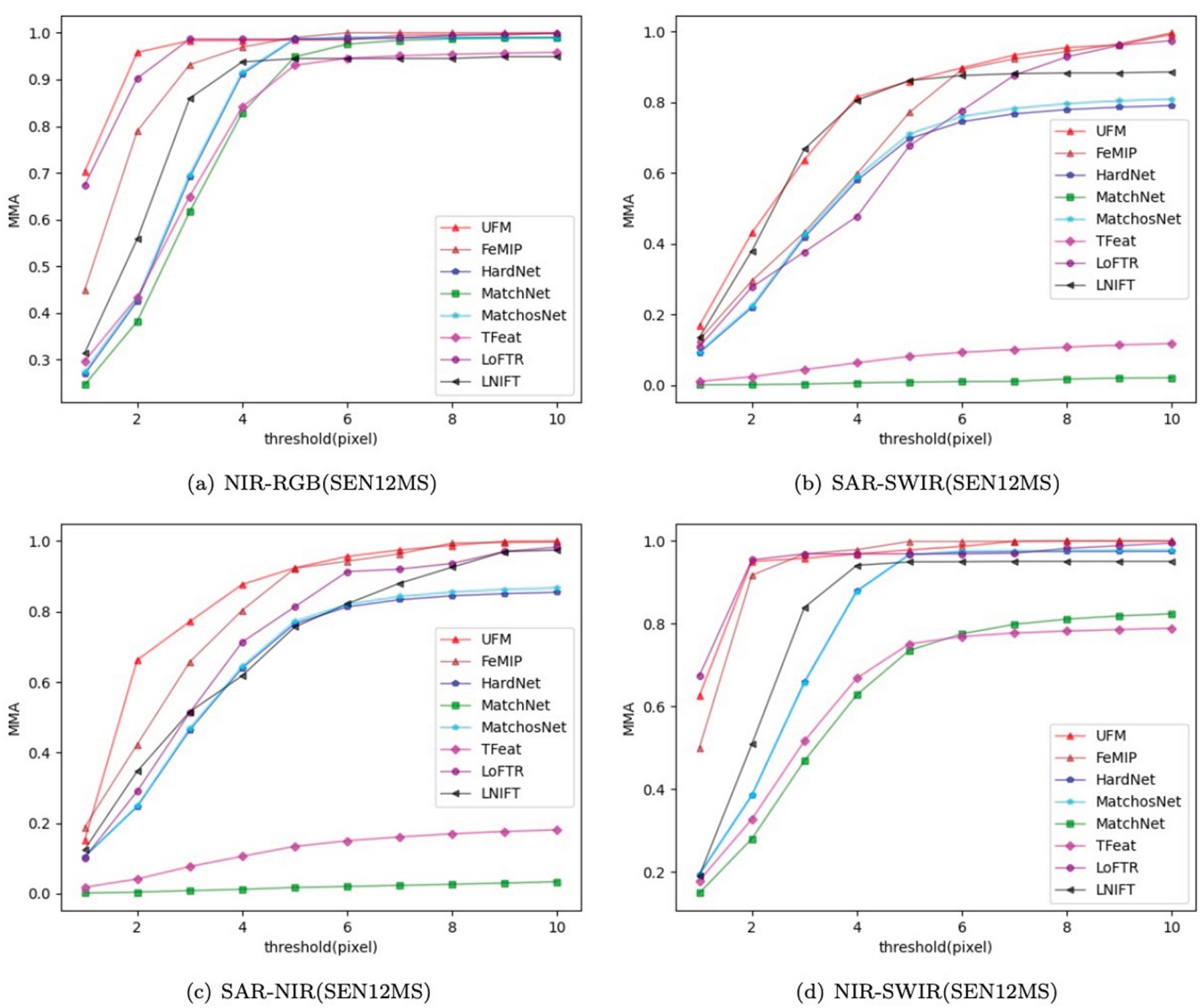

**Fig 9. MMA estimation on multi-modal images in SEN12MS dataset.**

points with horizontal and vertical pixel errors less than 1. The total error is computed based on both the horizontal and vertical pixel errors, and its value may exceed 1. In this experiment the horizontal distance error is defined as $H_{rmse}$, the vertical distance error is defined as $V_{rmse}$ and the total distance error on the image is defined as $HV_{rmse}$. All errors are measured in pixels. The functions $H_{rmse}$, $V_{rmse}$, and $HV_{rmse}$ are computed as:

$$H_{rmse} = \sqrt{\frac{1}{N} \sum_i \left( a_i^x - a_i^y \right)^2} \tag{13}$$

$$V_{rmse} = \sqrt{\frac{1}{N} \sum_i \left( b_i^x - b_i^y \right)^2} \tag{14}$$

**Table 4. Homography estimation on different modal images in the SEN12MS dataset.**

| Different Modals | Method | Homography est. AUC | | |
| --- | --- | --- | --- | --- |
| | | @3 px | @5 px | @10 px |
| NIR-RGB | MatchNet [81] | 29.49 | 49.61 | 71.85 |
| | Tfeat [82] | 45.22 | 60.28 | 75.31 |
| | LNIFT [24] | 53.07 | 66.22 | 78.79 |
| | HardNet [83] | 64.44 | 76.37 | 88.18 |
| | MatchosNet [68] | 63.64 | 77.32 | 87.92 |
| | LoFTR [27] | 63.69 | 76.71 | 87.69 |
| | FeMIP [18] | 61.30 | 77.93 | 88.21 |
| | UFM | **71.92** | **78.21** | **88.74** |
| SAR-SWIR | MatchNet [81] | 0.16 | 0.73 | 1.13 |
| | Tfeat [82] | 0.34 | 0.92 | 2.17 |
| | LNIFT [24] | 1.07 | 2.38 | 3.44 |
| | HardNet [83] | 7.33 | 20.85 | 42.54 |
| | MatchosNet [68] | 8.51 | 21.90 | 42.75 |
| | LoFTR [27] | 32.37 | 49.63 | 67.41 |
| | FeMIP [18] | 28.26 | 43.09 | 70.69 |
| | UFM | **33.01** | **49.72** | **70.88** |
| SAR-NIR | MatchNet [81] | 0.11 | 0.36 | 1.24 |
| | Tfeat [82] | 0.57 | 1.66 | 4.74 |
| | LNIFT [24] | 1.46 | 3.12 | 5.09 |
| | HardNet [83] | 10.08 | 24.77 | 47.90 |
| | MatchosNet [68] | 10.47 | 25.62 | 48.52 |
| | LoFTR [27] | 39.72 | 53.17 | 75.67 |
| | FeMIP [18] | 23.19 | 53.26 | 76.15 |
| | UFM | **39.84** | **57.81** | **76.19** |
| NIR-SWIR | MatchNet [81] | 10.87 | 21.51 | 37.28 |
| | Tfeat [82] | 29.03 | 42.60 | 56.06 |
| | LNIFT [24] | 49.10 | 65.04 | 78.99 |
| | HardNet [83] | 51.54 | 68.85 | 82.94 |
| | MatchosNet [68] | 50.48 | 68.19 | 82.56 |
| | LoFTR [27] | 61.98 | 75.50 | 86.97 |
| | FeMIP [18] | 66.45 | 79.77 | 89.89 |
| | UFM | **66.51** | **79.84** | **90.03** |

$$HV_{rmse} = \sqrt{\frac{1}{N} \sum_i \left( \left(a_i^x - a_i^y\right)^2 + \left(b_i^x - b_i^y\right)^2 \right)} \tag{15}$$

where $(a_i^x, b_i^x)$ denotes the coordinates of the matching points in the X-modal image, $(a_i^y, b_i^y)$ indicates the coordinates of the matching point in the Y-modal image. N denotes the total number of matched points.

As shown in Table 6, UFM consistently achieves the smallest distance error in most cases, outperforming other methods. The experiments further demonstrate that UFM not only identifies the maximum number of matching points with an error of less than 1 pixel, but also ensures that the error of the obtained matching points remains very small.

## 4.6 Evaluation of computational cost

To objectively evaluate the computational cost of the UFM algorithm, we compare the matching speed and resource requirements across different methods. In this experiment, the matching methods are tested on the RGB-NIR Scene dataset using an RTX 3090 GPU and an Intel i7-11700 processor. Matching speed is measured in frames per second (FPS), and resource usage is assessed in terms of storage requirements (MB). A higher FPS indicates faster inference, while lower storage values reflect reduced resource usage. As shown in Table 7, UFM

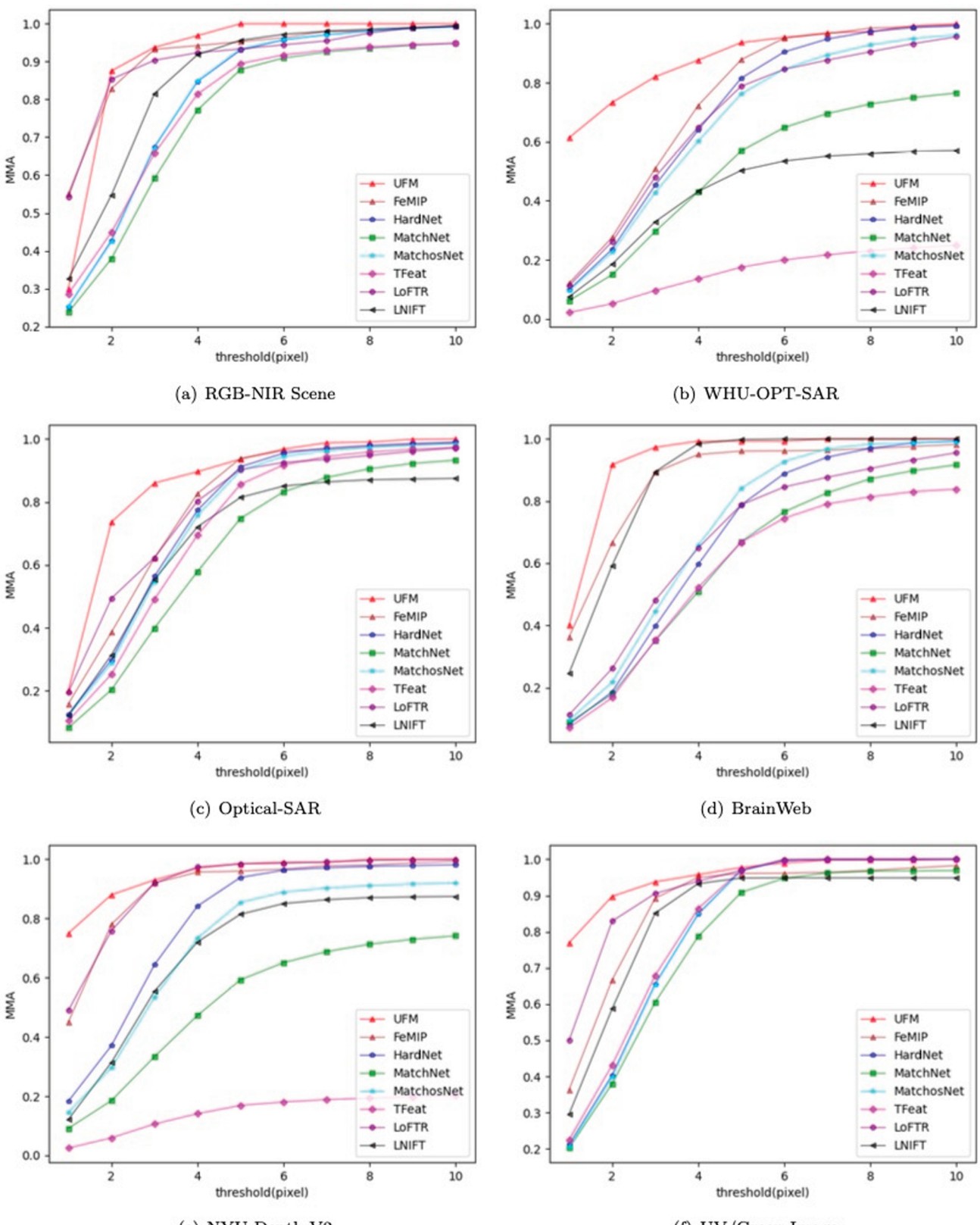

**Fig 10. MMA estimation on different multi-modal image datasets.** The modal of the images: (1) RGB-NIR, (2) Optical-SAR, (3) Optical-SAR, (4) T1-T2 (5) RGB-Depth, (6) UV-Green.

**Table 5. Homography estimation on different datasets.**

| Different modals | Method | Homography est. AUC | | |
|---|---|---|---|---|
| | | @3 px | @5 px | @10 px |
| RGB-NIR Scene | MatchNet [81] | 29.51 | 46.00 | 65.14 |
| | Tfeat [82] | 38.01 | 53.13 | 71.64 |
| | LNIFT [24] | 41.10 | 57.21 | 77.46 |
| | HardNet [83] | 39.03 | 54.59 | 73.38 |
| | MatchosNet [68] | 39.55 | 55.90 | 74.63 |
| | LoFTR [27] | 41.66 | 60.66 | 78.26 |
| | FeMIP [18] | 37.56 | 61.74 | 78.65 |
| | UFM | **42.22** | **64.20** | **82.10** |
| Brainweb | MatchNet [81] | 2.75 | 6.64 | 30.40 |
| | Tfeat [82] | 1.88 | 3.91 | 21.40 |
| | LNIFT [24] | 5.93 | 23.70 | 52.81 |
| | HardNet [83] | 3.67 | 8.85 | 39.23 |
| | MatchosNet [68] | 4.49 | 16.76 | 43.65 |
| | LoFTR [27] | 48.92 | 65.87 | 81.02 |
| | FeMIP [18] | 49.80 | 66.42 | 81.46 |
| | UFM | **49.91** | **66.49** | **83.27** |
| WHU-OPT-SAR | MatchNet [81] | 3.83 | 14.39 | 32.46 |
| | Tfeat [82] | 2.52 | 13.47 | 30.06 |
| | LNIFT [24] | 8.63 | 20.12 | 35.97 |
| | HardNet [83] | 16.51 | 39.10 | 66.25 |
| | MatchosNet [68] | 14.72 | 33.20 | 58.81 |
| | LoFTR [27] | 17.40 | 41.98 | 67.57 |
| | FeMIP [18] | 21.72 | 44.70 | 69.75 |
| | UFM | **33.02** | **48.76** | **72.06** |
| NYU-Depth V2 | MatchNet [81] | 0.61 | 3.51 | 15.35 |
| | Tfeat [82] | 0.42 | 2.39 | 12.33 |
| | LNIFT [24] | 11.19 | 22.67 | 36.02 |
| | HardNet [83] | 18.35 | 39.02 | 63.37 |
| | MatchosNet [68] | 12.54 | 28.80 | 52.09 |
| | LoFTR [27] | 61.89 | 76.04 | 88.03 |
| | FeMIP [18] | 69.08 | 80.36 | 89.72 |
| | UFM | **75.07** | **80.38** | **89.81** |
| Optical-SAR | MatchNet [81] | 3.78 | 14.24 | 38.04 |
| | Tfeat [82] | 12.74 | 31.06 | 56.39 |
| | LNIFT [24] | 15.47 | 36.06 | 59.66 |
| | HardNet [83] | 24.19 | 45.27 | 67.89 |
| | MatchosNet [68] | 20.87 | 42.36 | 65.38 |
| | LoFTR [27] | 29.54 | 45.66 | 64.86 |
| | FeMIP [18] | 28.89 | 49.40 | 70.07 |
| | UFM | **38.16** | **57.61** | **74.75** |
| UV-Green | MatchNet [81] | 27.48 | 37.98 | 56.88 |
| | Tfeat [82] | 31.23 | 50.21 | 74.38 |
| | LNIFT [24] | 23.44 | 37.10 | 60.42 |
| | HardNet [83] | 38.16 | 58.97 | 79.49 |
| | MatchosNet [68] | 41.86 | 63.25 | 81.66 |
| | LoFTR [27] | 62.38 | 77.01 | 87.41 |
| | FeMIP [18] | 66.06 | 77.29 | 88.52 |
| | UFM | **68.99** | **79.94** | **89.97** |

achieves the fastest matching speed. Among the methods compared, UFM, LoFTR, and FeMIP are semi-dense matching techniques, while the remaining methods are sparse matching approaches. Generally, semi-dense methods offer higher matching accuracy but tend to demand more resources. Notably, UFM requires the least resources among the semi-dense methods.

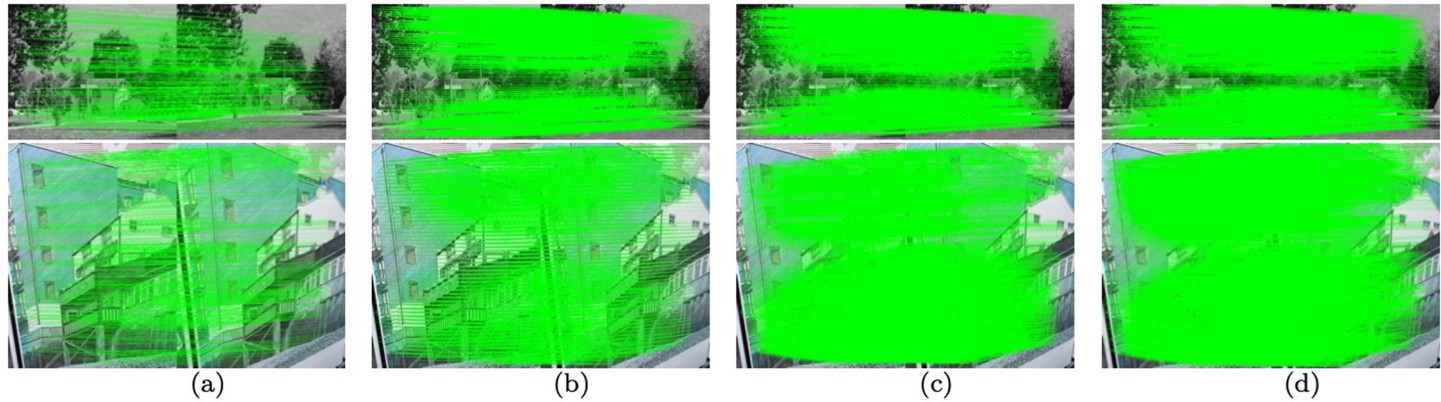

**Fig 11. Feature matching of same-modal images.** (a) SuperGlue (b) MatchFomer (c) LoFTR (d) UFM. The matches with less than 1 pixel error are represented by lines.

**Fig 12. Feature matching of different-modals images: (a) MatchosNet (b) LoFTR (c) FeMIP, and (d) UFM.** The datasets from top to bottom are: SEN 12 MS dataset, RGB-NIR Scene dataset, WHU-OPT-SAR dataset, Optical-SAR dataset, BrainWeb dataset, NYU-Depth V2 dataset and UV-Green dataset. The images of SEN 12 MS dataset, RGB-NIR Scene dataset and WHU-OPT-SAR dataset are rotated and the images of the Optical-SAR dataset and the NYU-Depth V2 dataset are cropped. The matches with less than 1 pixel error are represented by lines.

**Table 6. The distance error ($H_{rmse}$, $V_{rmse}$, $HV_{rmse}$) test of different methods on different datasets.**

| Method | Brainweb | NYU-Depth V2 | RGB-NIR Scene | WHU-OPT-SAR | UV-Green | SEN12MS |
|---|---|---|---|---|---|---|
| MatchNet [81] | 0.77,0.78,1.09 | 0.79,0.78,1.11 | 0.87,0.67,1.23 | 0.82,0.89,1.08 | 0.87,0.80,1.23 | 0.96,0.97,1.12 |
| Tfeat [82] | 0.79,0.78,1.12 | 0.75,0.89,1.07 | 0.68,0.65,0.96 | 0.79,0.81,1.12 | 0.85,0.84,1.16 | 0.65,0.62,0.89 |
| HardNet [83] | 0.79,0.80,1.12 | 0.77,0.76,1.08 | 0.70,0.69,0.99 | 0.81,0.81,0.98 | 0.75,0.76,1.07 | 0.64,0.65,0.91 |
| LNIFT [24] | 0.78,0.77,1.14 | 0.79,0.68,1.05 | 0.67,0.66,0.93 | 0.77,0.79,1.03 | 0.76,0.74,1.08 | 0.64,0.64,0.90 |
| MatchosNet [68] | 0.77,0.79,1.10 | 0.77,0.75,1.09 | 0.68,0.64,0.96 | 0.80,0.79,1.04 | 0.74,0.73,1.05 | 0.63,0.65,0.87 |
| LoFTR [27] | 0.60,0.68,0.92 | 0.72,0.66,1.02 | 0.75,0.89,1.07 | 0.82,0.89,1.08 | 0.69,0.72,0.96 | 0.64,**0.60**,**0.81** |
| FeMIP [18] | 0.55,**0.58**,0.82 | 0.67,0.51,0.96 | **0.63**,0.64,0.91 | 0.69,0.67,0.80 | **0.66**,0.70,0.98 | 0.59,0.67,0.82 |
| UFM | **0.53**,0.60,**0.76** | **0.32**,**0.29**,**0.45** | **0.63**,**0.62**,**0.89** | **0.50**,**0.56**,**0.72** | 0.67,**0.61**,**0.94** | **0.54**,0.68,**0.81** |

**Table 7. Evaluation of matching speed and resources on the RGB-NIR Scene dataset.**

| Method | Frames Per Second (FPS) ↑ | Storage (MB) ↓ |
|---|---|---|
| MatchNet [81] | 1.49 | 2.976 |
| Tfeat [82] | 1.17 | **0.334** |
| HardNet [83] | 0.56 | 1.283 |
| MatchosNet [68] | 1.30 | 0.368 |
| LoFTR [27] | 3.59 | 34.780 |
| FeMIP [18] | 4.80 | 34.782 |
| UFM [18] | **13.56** | 20.666 |

**Table 8. Ablation experiments on feature matching of same-modal images. A.Data Augmentation, B.Generic FFN, C.Assistant FFN, D.Pre-training, E.Fine-tuning. (1)–(7) are seven variants of UFM. (8) is the full UFM.**

| Method | A | MIA transformer | | D | E | Overall | Illumination | Viewpoint |
|---|---|---|---|---|---|---|---|---|
| | | B | C | | | Accuracy($\%, \epsilon < 1/3/5px$) | | |
| (1) | ✓ | ✗ | ✗ | ✓ | ✓ | 0.28 / 0.53 / 0.61 | 0.40 / 0.71 / 0.79 | 0.09 / 0.38 / 0.54 |
| (2) | ✗ | ✓ | ✗ | ✓ | ✓ | 0.44 / 0.69 / 0.81 | 0.67 / 0.88 / 0.90 | 0.12 / 0.45 / 0.58 |
| (3) | ✗ | ✗ | ✓ | ✓ | ✓ | 0.43 / 0.71 / 0.82 | 0.64 / 0.85 / 0.91 | 0.19 / 0.46 / 0.57 |
| (4) | ✓ | ✓ | ✗ | ✓ | ✓ | 0.52 / 0.79 / 0.84 | 0.79 / 0.95 / 0.96 | 0.32 / 0.63 / 0.74 |
| (5) | ✓ | ✗ | ✓ | ✓ | ✓ | 0.53 / 0.79 / 0.85 | 0.80 / 0.96 / 0.98 | 0.33 / 0.62 / 0.73 |
| (6) | ✓ | ✓ | ✓ | ✓ | ✗ | 0.49 / 0.74 / 0.76 | 0.78 / 0.91 / 0.92 | 0.29 / 0.60 / 0.67 |
| (7) | ✓ | ✓ | ✓ | ✗ | ✓ | 0.54 / 0.80 / 0.86 | 0.82 / 0.98 / 0.99 | 0.35 / 0.67 / 0.76 |
| (8) | ✓ | ✓ | ✓ | ✓ | ✓ | 0.56 / 0.82 / 0.87 | 0.83 / 0.99 / 0.99 | 0.37 / 0.69 / 0.77 |

## 4.7 Ablation study

To fully assess the role of different modules in UFM, seven variants were designed. In the 7th variant, pre-training is omitted, and the model is trained using the entire training dataset. As shown in Tables 8 and 9, we conduct ablation experiments on feature matching using images from the same modal and from different modals, respectively. The results of these experiments demonstrate that UFM outperforms all variants in feature matching. The combination of data augmentation and the MIA transformer is particularly effective, with the Assistant FFN in the MIA transformer having the most significant impact. The Generic FFN has a lesser effect on the matching performance. Data augmentation, Assistant FFN, and fine-tuning have a substantial influence on feature matching across different modals, while their impact on same-modal feature matching is relatively smaller. Although the 7th variant is trained on the full dataset, it still underperforms compared to the complete UFM, highlighting the necessity of the pre-training and fine-tuning mechanisms we have designed.

**Table 9. Ablation experiments on feature matching of different-modal images. A. Data Augmentation, B.Generic FFN, C.Assistant FFN, D.Pre-training, E.Fine-tuning. (1)–(7) are seven variants of UFM. (8) is the full UFM.**

| Method | A | MIA transformer | | D | E | Homography est. AUC | | |
|---|---|---|---|---|---|---|---|---|
| | | B | C | | | @3 px | @5 px | @10 px |
| (1) | ✓ | ✗ | ✗ | ✓ | ✓ | 39.88 | 53.44 | 62.01 |
| (2) | ✗ | ✓ | ✗ | ✓ | ✓ | 46.03 | 62.85 | 70.91 |
| (3) | ✗ | ✗ | ✓ | ✓ | ✓ | 58.71 | 66.02 | 76.57 |
| (4) | ✓ | ✓ | ✗ | ✓ | ✓ | 62.02 | 70.77 | 81.26 |
| (5) | ✓ | ✗ | ✓ | ✓ | ✓ | 70.88 | 77.94 | 88.09 |
| (6) | ✓ | ✓ | ✓ | ✓ | ✗ | 64.71 | 72.68 | 84.97 |
| (7) | ✓ | ✓ | ✓ | ✗ | ✓ | 71.13 | 77.98 | 88.53 |
| (8) | ✓ | ✓ | ✓ | ✓ | ✓ | 71.92 | 78.21 | 88.74 |

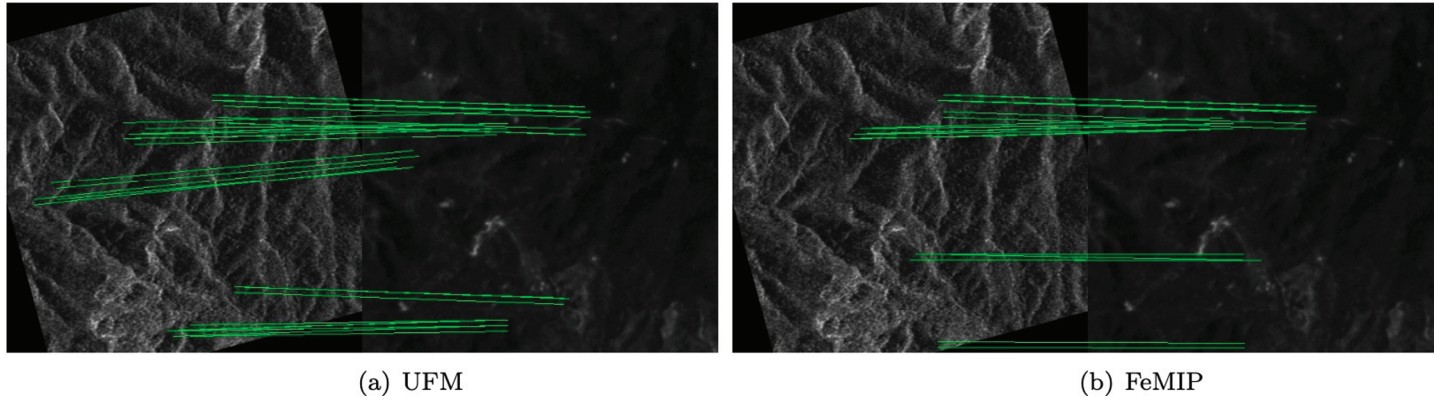

(a) UFM                                          (b) FeMIP

**Fig 13. Feature matching on images with very little texture.**

## 4.8 Limitation

Although UFM has been shown to deliver good matching results in most cases, it also has some limitations. As illustrated in Fig 13, UFM struggles to achieve accurate matching when dealing with multi-modal images that have very few textures. This is a common challenge for feature matching methods, and other approaches also face difficulties in such scenarios. While UFM has proven to be highly generalizable, its pre-training becomes less effective when confronted with unseen modal images. In such cases, as with other methods, it is necessary to train the model on the entire dataset to achieve accurate matching results.

## 5 Conclusion

This paper introduces UFM, a Unified Feature Matching model designed for fine-tuning across a broad range of modal images using a shared Multimodal Image Assistant (MIA) Transformer. MIA Transformers, serving as multimodal assistants, enhance a generic Feed-forward Network (FFN) by encoding modal-specific information. The shared self-attention and cross-attention mechanisms enable improved feature matching across different modals. Through the incorporation of data augmentation and staged pre-training, UFM demonstrates significantly enhanced pre-training effectiveness on multimodal images. Experimental results showcase UFM's capability to achieve excellent performance in diverse feature matching tasks.

Future work aims to enhance UFM further by expanding the pre-training dataset and integrating image data from less common modals. Consideration is given to transitioning from a

semi-dense matching framework to a dense matching framework to elevate matching accuracy. Plans also include scaling up the model size used in UFM pretraining. Additionally, ongoing research explores downstream tasks of image feature matching, with an emphasis on integrating the unified feature matching algorithm into diverse applications.

## Author contributions

**Conceptualization:** Yide Di, Yun Liao.

**Data curation:** Qing Duan, Junhui Liu.

**Formal analysis:** Yide Di, Yun Liao.

**Funding acquisition:** Qing Duan, Mingyu Lu.

**Investigation:** Qing Duan, Junhui Liu.

**Methodology:** Yun Liao, Hao Zhou.

**Project administration:** Yun Liao.

**Software:** Hao Zhou, Kaijun Zhu.

**Supervision:** Mingyu Lu.

**Validation:** Hao Zhou, Kaijun Zhu.

**Visualization:** Hao Zhou, Kaijun Zhu.

**Writing – original draft:** Yide Di.

**Writing – review & editing:** Yide Di.

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
