## [Decision Letter · Decision Letter 0]

15 Nov 2024

PONE-D-24-47780UFM: Unified Feature Matching Pre-training with Multi-Modal Image AssistantsPLOS ONE

Dear Dr. Liao,

Thank you for submitting your manuscript to PLOS ONE. After careful consideration, we feel that it has merit but does not fully meet PLOS ONE’s publication criteria as it currently stands. Therefore, we invite you to submit a revised version of the manuscript that addresses the points raised during the review process.

**ACADEMIC EDITOR: **Before we consider accepting this manuscript for publication, you must fully address the concerns raised by Reviewers 1, 2 and 3.** **

Reviewer #1: This paper introduces UFM (Unified Feature Matching), a pre-trained model designed to handle feature matching across multiple image modalities. The key innovation is the Multi-Modal Image Assistant (MIA) Transformer architecture that enables both same-modality and cross-modality feature matching through a shared parameter space. The authors propose a staged pre-training strategy and data augmentation techniques to address challenges with imbalanced modal datasets. The work is evaluated across various datasets including optical, SAR, NIR, SWIR, and medical images, demonstrating competitive performance against specialized methods.

Strengths

-Novel MIA Transformer architecture that effectively handles both same-modality and cross-modality matching

-Good solution to the modal imbalance problem through staged pre-training

Comprehensive Evaluation

-Addresses a real-world need for unified feature matching across modalities

Major issues:

-Data augmentation strategy needs more detailed explanation

-Transition between the pre-training stages is not clearly described

-Need for more extensive failure cases and analysis to help readers know the limitation

Presentation Issues

-Figures captions are mostly lack sufficient details and explanations.

-How to address with or make sure the training stability across different modalities?

-Few past works in feature matching are missing in literature review, including [Maskflownet: Asymmetric feature matching with learnable occlusion mask][Superthermal: Matching thermal as visible through thermal feature exploration][CorMatcher: A corners-guided graph neural network for local feature matching], etc

Reviewer #2: - Page 3, contribution 3: Please revise the sentence "Remarkably, it has exhibited remarkable performance."

- Method description could benefit from more clarity in the architecture overview. At times, it is not clear to me which components relate to the UFM and which to the MIA. This could be reinforced with small augmentations to Fig. 2.

- Method could also use clarity on the nature of the model's outputs. There's a per-element mapping matrix (GT_matrix) which is more or less described, but there are also references to a "heatmap" which first appears on page 8 in discussion of the loss function.

- Given the non-linear intensity relationships across imaging modalities, I find it curious that all data augmentations are purely geometric. Were any intensity augmentations considered?

- I am slightly unclear about the training sequence for the feature assistants. The presented training steps seem to be three stages of pretraining: 1) Generic FFN Pre-training, 2) train all X-X assistants, 3) train all X-Y assistants. If this is correct, then I think the text and Fig. 4 could clarify that these are three stages of pretraining. Since the example only considers two modalities (Opt, SAR), how does this method change for greater numbers of modalities? It seems like all X-X assistants can be trained in parallel, but that all X-Y assistants might need to be trained simultaneously.

- The experiments do not describe when fine-tuning was deployed and what data was used for model fine-tuning.

- It's not clear to me how Homography estimation is a valuable tasks for the different-modals same-scene feature matching. My understanding was that these image pairs are already close to alignment, as shown in Fig. 8. Is some augmentation imposed?

- In the beginning of sec 4.4, you simply state that "the correct matches are represented by lines," but do not define a "correct match." What is the error margin for a "correct match"? Later, you state that it's <1px error, but not here.

- Fig. 11 and Fig. 12 - Please add more description to the caption, specifically mentioning that only matches with <1px error are plotted.

- For Fig 12, it actually isn't visually clear that your method is superior. Perhaps consider a plot of the number of matches with <1px error across methods?

Reviewer #3: The authors propose a UFM model, leveraging MIA Transformers, a novel approach to unified feature matching across diverse modalities.

Strengths: The experiments are very extensive by including a set of datasets and comparisons with state-of-the-art methods. The methodology is detailed in explaining the architecture, pre-training strategies, and data augmentation. Finally, the results, especially in comparison with existing methods, substantiate the model's effectiveness and generalizability.

Weaknesses:

1) fig 5 and fig 6 could be improved with correcting the subscripts and defining what '...' means or in what area is including the '...'

2) while the performance gains are evident, I would like to see a statistical analysis to confirm the significance observed differences, such as p-values or confidence intervals.

3) the absence of variability measures makes it difficult to assess the consistency and robustness of results. Such as, standard deviations or error bars.

4) There is little discussion on the computational cost of the UFM model compared to competing methods. Including this would give a more complete picture of its practical applicability

5) Quantitative are strong, but there is a lack of qualitative evaluation of feature matching performances. More visual examples would help.

6) While UFM is presented as highly generalizable, more empirical evidence or theoretical discussion on its adaptability to less common modalities would strengthen this claim.

7) while the ablation studies are valuable, deeper interpretation of the interactions between the model components (data augmentation and MIA transformers interaction) to determine the performance boost. 

We look forward to receiving your revised manuscript.

Kind regards,

Ziyu Qi

Academic Editor

PLOS ONE

Journal Requirements:

“This work was supported in part by the National Natural Science Foundation of China under Grant 61976124 and 62372077. This work was also supported in part by the Scientific Research Fund of Yunnan Provincial Education Department under Grant 2021J0007.”

Additional Editor Comments (if provided):

Before we consider accepting this manuscript for publication, you must fully address the concerns raised by Reviewers 1, 2 and 3.

Reviewers' comments:

Reviewer's Responses to Questions

**Comments to the Author**

1. Is the manuscript technically sound, and do the data support the conclusions?

Reviewer #1: Yes

Reviewer #2: Partly

Reviewer #3: Yes

2. Has the statistical analysis been performed appropriately and rigorously? 

Reviewer #1: Yes

Reviewer #2: I Don't Know

Reviewer #3: Yes

3. Have the authors made all data underlying the findings in their manuscript fully available?

Reviewer #1: Yes

Reviewer #2: Yes

Reviewer #3: Yes

4. Is the manuscript presented in an intelligible fashion and written in standard English?

Reviewer #1: Yes

Reviewer #2: Yes

Reviewer #3: Yes

5. Review Comments to the Author

Reviewer #1: This paper introduces UFM (Unified Feature Matching), a pre-trained model designed to handle feature matching across multiple image modalities. The key innovation is the Multi-Modal Image Assistant (MIA) Transformer architecture that enables both same-modality and cross-modality feature matching through a shared parameter space. The authors propose a staged pre-training strategy and data augmentation techniques to address challenges with imbalanced modal datasets. The work is evaluated across various datasets including optical, SAR, NIR, SWIR, and medical images, demonstrating competitive performance against specialized methods.

Strengths

-Novel MIA Transformer architecture that effectively handles both same-modality and cross-modality matching

-Good solution to the modal imbalance problem through staged pre-training

Comprehensive Evaluation

-Addresses a real-world need for unified feature matching across modalities

Major issues:

-Data augmentation strategy needs more detailed explanation

-Transition between the pre-training stages is not clearly described

-Need for more extensive failure cases and analysis to help readers know the limitation

Presentation Issues

-Figures captions are mostly lack sufficient details and explanations.

-How to address with or make sure the training stability across different modalities?

-Few past works in feature matching are missing in literature review, including [Maskflownet: Asymmetric feature matching with learnable occlusion mask][Superthermal: Matching thermal as visible through thermal feature exploration][CorMatcher: A corners-guided graph neural network for local feature matching], etc

Reviewer #2: - Page 3, contribution 3: Please revise the sentence "Remarkably, it has exhibited remarkable performance."

- Method description could benefit from more clarity in the architecture overview. At times, it is not clear to me which components relate to the UFM and which to the MIA. This could be reinforced with small augmentations to Fig. 2.

- Method could also use clarity on the nature of the model's outputs. There's a per-element mapping matrix (GT_matrix) which is more or less described, but there are also references to a "heatmap" which first appears on page 8 in discussion of the loss function.

- Given the non-linear intensity relationships across imaging modalities, I find it curious that all data augmentations are purely geometric. Were any intensity augmentations considered?

- I am slightly unclear about the training sequence for the feature assistants. The presented training steps seem to be three stages of pretraining: 1) Generic FFN Pre-training, 2) train all X-X assistants, 3) train all X-Y assistants. If this is correct, then I think the text and Fig. 4 could clarify that these are three stages of pretraining. Since the example only considers two modalities (Opt, SAR), how does this method change for greater numbers of modalities? It seems like all X-X assistants can be trained in parallel, but that all X-Y assistants might need to be trained simultaneously.

- The experiments do not describe when fine-tuning was deployed and what data was used for model fine-tuning.

- It's not clear to me how Homography estimation is a valuable tasks for the different-modals same-scene feature matching. My understanding was that these image pairs are already close to alignment, as shown in Fig. 8. Is some augmentation imposed?

- In the beginning of sec 4.4, you simply state that "the correct matches are represented by lines," but do not define a "correct match." What is the error margin for a "correct match"? Later, you state that it's <1px error, but not here.

- Fig. 11 and Fig. 12 - Please add more description to the caption, specifically mentioning that only matches with <1px error are plotted.

- For Fig 12, it actually isn't visually clear that your method is superior. Perhaps consider a plot of the number of matches with <1px error across methods?

Reviewer #3: The authors propose a UFM model, leveraging MIA Transformers, a novel approach to unified feature matching across diverse modalities.

Strengths: The experiments are very extensive by including a set of datasets and comparisons with state-of-the-art methods. The methodology is detailed in explaining the architecture, pre-training strategies, and data augmentation. Finally, the results, especially in comparison with existing methods, substantiate the model's effectiveness and generalizability.

Weaknesses:

1) fig 5 and fig 6 could be improved with correcting the subscripts and defining what '...' means or in what area is including the '...'

2) while the performance gains are evident, I would like to see a statistical analysis to confirm the significance observed differences, such as p-values or confidence intervals.

3) the absence of variability measures makes it difficult to assess the consistency and robustness of results. Such as, standard deviations or error bars.

4) There is little discussion on the computational cost of the UFM model compared to competing methods. Including this would give a more complete picture of its practical applicability

5) Quantitative are strong, but there is a lack of qualitative evaluation of feature matching performances. More visual examples would help.

6) While UFM is presented as highly generalizable, more empirical evidence or theoretical discussion on its adaptability to less common modalities would strengthen this claim.

7) while the ablation studies are valuable, deeper interpretation of the interactions between the model components (data augmentation and MIA transformers interaction) to determine the performance boost.

6. PLOS authors have the option to publish the peer review history of their article (what does this mean?). If published, this will include your full peer review and any attached files.

Reviewer #1: No

Reviewer #2: No

Reviewer #3: **Yes: **Rubi Quiñones

---

## [Author Response · Author response to Decision Letter 1]

20 Dec 2024

Thank you very much for your suggestions. Since there are so many replies, I have put all the replies in the file "Response to Editors and Reviewers Comments_2024_12_20". We do appreciate your serious consideration for our corrections in this version.

---

## [Decision Letter · Decision Letter 1]

27 Jan 2025

UFM: Unified Feature Matching Pre-training with Multi-Modal Image Assistants

PONE-D-24-47780R1

Dear Dr. Liao,

We’re pleased to inform you that your manuscript has been judged scientifically suitable for publication and will be formally accepted for publication once it meets all outstanding technical requirements.

Kind regards,

Paulo Eduardo Teodoro, Dr.

Academic Editor

PLOS ONE

Additional Editor Comments (optional):

Reviewers' comments:

Reviewer's Responses to Questions

**Comments to the Author**

1. If the authors have adequately addressed your comments raised in a previous round of review and you feel that this manuscript is now acceptable for publication, you may indicate that here to bypass the “Comments to the Author” section, enter your conflict of interest statement in the “Confidential to Editor” section, and submit your "Accept" recommendation.

Reviewer #1: All comments have been addressed

Reviewer #3: All comments have been addressed

2. Is the manuscript technically sound, and do the data support the conclusions?

Reviewer #1: Yes

Reviewer #3: Yes

3. Has the statistical analysis been performed appropriately and rigorously? 

Reviewer #1: Yes

Reviewer #3: Yes

4. Have the authors made all data underlying the findings in their manuscript fully available?

Reviewer #1: Yes

Reviewer #3: Yes

5. Is the manuscript presented in an intelligible fashion and written in standard English?

Reviewer #1: Yes

Reviewer #3: Yes

6. Review Comments to the Author

Reviewer #1: This paper introduces UFM (Unified Feature Matching), a pre-trained model designed to handle feature matching across multiple image modalities. The key innovation is the Multi-Modal Image Assistant (MIA) Transformer architecture that enables both same-modality and cross-modality feature matching through a shared parameter space.

As a revised submission, the reviewer has no further new comments to add. Thanks for authors' efforts!

Reviewer #3: (No Response)

7. PLOS authors have the option to publish the peer review history of their article (what does this mean?). If published, this will include your full peer review and any attached files.

Reviewer #1: No

Reviewer #3: **Yes: **Dr. Rubi Quiñones

---

## [Editor Report · Acceptance letter]

PONE-D-24-47780R1

PLOS ONE

Dear Dr. Liao,

I'm pleased to inform you that your manuscript has been deemed suitable for publication in PLOS ONE. Congratulations! Your manuscript is now being handed over to our production team.

Kind regards,

on behalf of

Professor Paulo Eduardo Teodoro

Academic Editor

PLOS ONE